# Women’s Media Use and Preferences of Media-Based Interventions on Lifestyle-Related Risk Factors in Gynecological and Obstetric Care: A Cross-Sectional Multi-Center Study in Germany

**DOI:** 10.3390/ijerph18189840

**Published:** 2021-09-18

**Authors:** Manuela Bombana, Maren Wittek, Gerhard Müller, Monika Heinzel-Gutenbrunner, Michel Wensing

**Affiliations:** 1Department of General Practice and Health Service Research, Heidelberg University Hospital, Im Neuenheimer Feld 130.3, 69120 Heidelberg, Germany; maren.wittek@gero.uni-heidelberg.de (M.W.); michel.wensing@med.uni-heidelberg.de (M.W.); 2Department of Health Promotion, AOK Baden-Württemberg, Presselstrasse 19, 70191 Stuttgart, Germany; gerhard.mueller@bw.aok.de; 3MH Statistics Consulting, Bienenweg 8, 35041 Marburg, Germany; Monika.Heinzel@mh-statistik.com

**Keywords:** media-based interventions, gynecological and obstetric care, health-related behaviors, pregnancy, lactation

## Abstract

This study aimed to investigate factors affecting (1) women’s media use regarding health-related behaviors during pregnancy and lactation, (2) women’s preferences for media format, and (3) the content of media-based interventions on lifestyle-related risk factors during pregnancy and lactation. A cross-sectional observational multi-center study of pregnant and lactating women and women of childbearing age was carried out in 14 randomly selected obstetric and gynecologic care settings in the 12 most populated cities in Baden-Wuerttemberg, South-West Germany. Data from 219 surveyed women showed that older women, pregnant women, and lactating women have a higher probability of using media during pregnancy and lactation, respectively. The majority of women preferred a combination of analog and digital media-based interventions in gynecological (46.9%) and obstetric (47.1%) care settings and at home (73.0%). Women would like to see information brochures and flyers on health-related behaviors during pregnancy and lactation for use in gynecological and obstetric care settings, and for media use at home, they would like to have books. The probability of preferring the favored media formats in gynecological and obstetric care settings and at home were associated with pregnancy status, relationship status, socioeconomic status (SES), ethnicity, and health insurance status. About 80% of the surveyed women preferred media content regarding recommendations for a healthy lifestyle and healthy behavior during pregnancy and lactation. All of the independent variables were associated with the probability of preferring a specific media content. The SES was found to play a major role in the probability of preferring a specific media content, followed by pregnancy status, ethnicity, and health insurance status. The results from our study provide a basis for tailored preventive interventions in gynecological and obstetric care settings and for use at home. The results imply that a woman can be reached before conception, during pregnancy, or during lactation with preventive measures tailored to their requirements; however, acceptance may vary across personal attributes, such as SES, ethnicity, and others.

## 1. Introduction

Numerous scientific studies have investigated the effects of lifestyle-related risk factors (LRRFs) during pregnancy and lactation on the (unborn) child [1,2,3,4,5,6,7,8,9,10]. According to the WHO definition, risk factors are attributes, characteristics, or exposures of an individual that increase the risks of developing a disease or injury [11]. LRRFs during pregnancy and lactation include all factors that may cause premature death of the child due to non-communicable diseases as a result of an unfavorable maternal lifestyle [12,13]. These mainly include risks associated with substance use, nutrition, stress, medication, oral health, and physical inactivity [3,12,14]. The authors conducted comprehensive literature reviews on the effects of LRRFs in pregnancy and lactation and described the health effects of LRRFs on the offspring, prevalence rates, and the current recommendations in great detail [15,16]. In a cross-sectional study that was conducted in 2018, we surveyed 209 pregnant women in Germany. The study showed that there is a huge knowledge gap regarding LRRFs during pregnancy among pregnant women and an urgent need to educate women regarding LRRFs during pregnancy [17]. We found that gynecologists in particular seem effective in providing information on LRRFs during pregnancy [17]. To sustainably reduce LRRFs during pregnancy and lactation, the implementation of media-based educational interventions in gynecological and obstetric care settings and for women’s use at home are a possible approach to address LRRFs during pregnancy and lactation to protect offspring health.

In Germany, standardized and comprehensive education and information transfers on health-related behaviors during pregnancy and lactation have not yet been included in the existing gynecological and obstetric guidelines [18]. Accordingly, questioning and monitoring pregnant women’s lifestyle is not routinely applied in gynecological and obstetric care. However, the guidelines determine that the gynecologists need to provide information in case of alcohol, nicotine, drug use, or others during pregnancy [18]. There are currently no references to education about LRRFs during the breastfeeding period in the German guidelines [18]. According to information from the German Society of Gynecologists and Obstetricians, this guideline is revised at the moment.

For the development of tailored media-based interventions on LRRFs during pregnancy and lactation in gynecological and obstetric care settings and for women’s use at home, it is relevant to know how to reach and address the target groups.

Therefore, this study aimed to investigate factors affecting (1) women’s media use on health-related behaviors during pregnancy and lactation, (2) women’s preferences of format, and (3) the content of media-based interventions on LRRFs during pregnancy and lactation.

## 2. Materials and Methods

### 2.1. Study Design and Setting

This study was a cross-sectional observational multi-center study of pregnant and breastfeeding women as well as women of childbearing age in Baden-Wuerttemberg, Germany. A random selection of participating gynecological and obstetric care settings was computer generated. Health insurer AOK Baden-Wuerttemberg (BW) contacted 147 institutions of gynecological and obstetric care via mail or e-mail. Following recruitment, reminders were sent to e-mail contacts after one week of non-response and again after nine days of non-response. Reminder calls were made 14 days after initial contact for all institutions with non-response. A total of 14 of the contacted institutions granted permission to recruit and survey women in their gynecological or obstetric institution. Thus, we recruited eligible women in 14 different settings and institutions of gynecological and obstetric care in BW, Germany, from 1 October to 15 November 2019. The cities and the respective number of sample points are shown in Figure 1.

Of the 147 contacted institutions of gynecological and obstetric care, 14 agreed to participate in the study (response rate: 9.5%).

### 2.2. Study Population and Data Collection

The study population consisted of pregnant and lactating women as well as women of childbearing age (≤49 years). Only women of legal age (≥18 years of age) were authorized to participate in the survey. After the random selection of gynecological and obstetric institutions, eligible women were recruited in the following settings: waiting room, tour of deliver and maternity ward, prenatal classes, and parents’ evenings in hospitals. A total of 252 questionnaires were distributed to women meeting the inclusion criteria. A total of 32 women declined to participate, and 220 women filled in the questionnaire. One woman did not meet the inclusion criteria. A total of 219 women (87.3% response rate) were included into our study.

Data were collected anonymously to reduce potential social desirability bias. The participating pregnant and breastfeeding women as well as the participating women of childbearing age were informed verbally and in writing before the interview started. Informed consent was obtained by inserting the questionnaire into a box. Withdrawal from the survey was possible at any time and without giving reasons until the questionnaire was inserted into a box. Withdrawal was no longer possible after the questionnaire was inserted into a box. The dataset will be provided by the corresponding author on reasonable request.

### 2.3. Questionnaire

We did not identify any existing German validated questionnaire covering the topics of interest. Therefore, we developed a self-administered questionnaire (available on request from the corresponding author).

The German questionnaire consisted of 44 items (Appendix A). Multiple choice questions with single-option or multi-option answers were applied. The items covered topics on awareness, preferences, and barriers to and problems of media use in gynecological and obstetric care, as well as socio-demographics. The questionnaire was pre-tested by five women and adjusted accordingly. We performed 12 subsequent pilot tests. After six subsequent pilot-tests, we did not obtain any new information and reached saturation. The women completed the questionnaire within 15–20 min on average.

### 2.4. Dependent Variables

#### 2.4.1. Women’s Media Use

The dependent variables were women’s media use regarding health-related behaviors during pregnancy (yes or no) and women’s media use regarding health-related behaviors during lactation (yes or no). We asked women whether they use available media regarding health-related behaviors during pregnancy and lactation with dichotomous answer categories.

#### 2.4.2. Format Preferences of Media-Based Interventions

Women’s preferred media technology in gynecological and obstetric care settings and for women’s use at home was a further dependent variable. We asked the women: “What media technology with information on health behaviors during pregnancy and lactation would you like to see for on-site use in a gynecologist’s practice and/or midwife’s practice?” (the answer categories were separated for gynecologist’s practice and midwife’s practice) and “What media technology with health behavior information during pregnancy and lactation would you like for use at home?” The answer categories were “analog, digital, analog and digital, no media”.

Women’s preferred media format in gynecological and obstetric care settings and for women’s use at home were further dependent variables. We asked the women: “Which of the following media formats containing information on health behaviors during pregnancy and lactation would you like to see for use on-site in the gynecologist’s practice and/or midwife’s practice?” (the answer categories were separated for gynecologist’s practice and midwife’s practice) and “Which of the following media with information on health behaviors during pregnancy and lactation would you like for use at home?” The answer categories were addresses (yes, no), apps (yes, no), books (yes, no), flyers (yes, no), information brochures (yes, no), leaflets (yes, no), audio recording/podcasts (yes, no), video material (yes, no), magazines (yes, no), and no media (yes, no).

#### 2.4.3. Content Preferences for Media-Based Interventions

Further dependent variables were women’s preferred media content regarding health-related behaviors during pregnancy and lactation. We asked the women “What should be the focus of the pregnancy-related information?” and “What should be the focus of information on lactation?” The answer categories were a variety of media contents, including recommendations for healthy lifestyle/behaviors (yes, no), recommendations for the avoidance of lifestyle-related risks/behaviors (yes, no), information on potential lifestyle-related risks (yes, no), recipes for cooking (yes, no), movement exercises (yes, no), relaxation exercises (yes, no), potential alternatives to tobacco and alcohol (yes, no), recommendations for the avoidance of specific medications (yes, no), and recommendations for essential supplements (yes, no).

### 2.5. Independent Variables

The independent variables were as follows: currently pregnant (yes, no), currently lactating (yes, no), firm relationship (yes, no), age (in years), socioeconomic status by Winkler (low, middle, high), ethnicity (German, non-German), health insurance status (private, statutory, other/none) [19,20]. The socioeconomic status (SES) was measured using ‘Winkler’s index’, which is a widely used social class index based on the validated ‘Scheuch index’, which combines information on net income, basic and the vocational education, and profession [21,22]. The variable is derived from the main wage earner in the household and categorized into ‘high, middle, and low’. Further details of the measurement and classification can be found elsewhere [21,22].

As a proxy of ethnicity, we used country of birth of the individual and both parents. If at least one parent was born abroad, the person was considered as non-German. In cases of mixed origin, the mother’s country of birth prevailed [21].

### 2.6. Statistical Analyses

Descriptive statistics were applied to investigate the sample characteristics. We applied multivariate logistic regressions to investigate the associations between independent variables and (1) media use on health-related behaviors during pregnancy and lactation; (2) women’s preferences in selected media formats regarding health-related behaviors during pregnancy and lactation in gynecological and obstetric care and at home; (3) women’s preferred media content on health-related behaviors during pregnancy and lactation in gynecological and obstetric care and at home.

We conducted all the tests for 95% confidence with α = 0.05. Data were analyzed using IBM Statistical Package of Social Sciences (SPSS) for Windows, Version 26.0 (IMB Corp. Released 2019. Armonk, NY, USA: IBM).

The study was conducted in accordance with the Declaration of Helsinki. The study received ethical approval from the Ethics Committee of the Medical Faculty of the Ruprecht Karls University on 9 May 2019 (S-289/2019). Participants were informed that by completing and delivering the questionnaire, their approval to participate in this study would be confirmed.

## 3. Results

The sample characteristics are shown in Table 1.

In multivariate logistic regression analyses, we investigated the association between media use during pregnancy and lactation (Table 2), respectively, in relation to independent variables. Women who were not currently pregnant were significantly less likely to use media regarding health-related behaviors during pregnancy (OR 0.30, 95% CI 0.13, 0.67). With increasing age, the ORs of using media regarding health-related behaviors during pregnancy significantly increased (OR 1.11, 95% CI 1.02, 1.21). Similar results were identified for the association between media use during lactation and selected independent variables: women who were not currently lactating were significantly less likely to use media regarding health-related behaviors during lactation (OR 0.21, 95% CI 0.05, 0.98). With increasing age, the ORs of using media regarding health-related behaviors during lactation significantly increased (OR 1.08, 95% CI 1.00, 1.17).

Of the surveyed women, 46.9% preferred a combination of analog and digital media use in gynecological care; 47.1% preferred a combination of analog and digital media use in obstetric care; 73.0% preferred a combination of analog and digital media for use at home. In each setting, the use of analog media was more preferred than the use of digital media.

We further investigated women’s preferred media formats regarding health-related behaviors during pregnancy and lactation in gynecological and obstetric care settings and at home (see Figure 2). The women had the possibility of selecting multiple answers. In gynecological care, the women preferred information brochures and flyers on health-related behaviors during pregnancy and lactation. Similar results were found for media use in obstetric care. At home, the women preferred books on lifestyle-related risks during pregnancy and lactation.

We further investigated the probability of preferring the two most popular media formats in association with independent variables in order to understand how to reach specific target groups with media-based interventions on health-related behaviors during pregnancy and lactation in gynecological and obstetric care and for use at home (Table 3). In gynecological care, we found that preferences for information brochures regarding health-related behaviors during pregnancy and lactation decreased with increasing age (OR 0.92, 95% CI 0.86, 0.99). Non-German women, as compared to German women, were less likely to prefer information brochures on health-related behaviors during pregnancy and lactation (OR 0.42, 95% CI 0.18, 0.99). Women who were not insured with a private or statutory health insurance were less likely to prefer information brochures on health-related behaviors during pregnancy and lactation (OR 0.16, 95% CI 0.32, 0.78). In obstetric care, we found that the probability of preferring information brochures regarding health-related behaviors during pregnancy and lactation was significantly higher in non-pregnant women (OR 3.26, 95% CI 1.29, 8.20) as compared to pregnant women. Similar to the results of the preference for information brochures during pregnancy in gynecological care, the probability of preferring information brochures regarding health-related behaviors during pregnancy and lactation increased with decreasing age in obstetric care. The probability of preferring information brochures on health-related behaviors during pregnancy and lactation was significantly lower in non-Germans as compared to Germans (OR 0.38, 95% CI 0.16, 0.89) and in women who were not in a firm relationship (OR 0.23, 95% CI 0.06, 0.95). The probability of women preferring flyers on health-related behaviors during pregnancy and lactation in obstetric care was significantly higher in women with a middle SES (OR 2.21, 95% CI 1.05, 4.66) as compared to women with a high SES. For use at home, we found that the probability of preferring addresses for use during pregnancy and lactation regarding issues of health-related behaviors during pregnancy and lactation was significantly lower in women with a low (OR 0.20, 95% CI 0.06, 0.70) and middle SES (OR 0.29, 95% CI 0.13, 0.61) as compared to high-SES women. The OR of preferring addresses for use during pregnancy and lactation regarding issues of health-related behaviors during pregnancy and lactation in the home setting was significantly higher for women with a statutory health insurance status (OR 2.72, 95% CI 1.00, 7.45) than for privately insured women. The probability of preferring an app for health-related behaviors during pregnancy and lactation at home was significantly lower in non-pregnant (OR 0.44, 95% CI 0.20, 1.00) and non-German women (OR 0.28, 95% CI 0.12, 0.67). For use at home, we found that the probability of preferring an app was significantly higher in women with a statutory health insurance status (OR 2.73, 95% CI 1.03, 7.22) as compared to privately insured women.

We asked women what the focus of media content for pregnancy and lactation should be. A total of 80.1% of the surveyed women preferred media content regarding recommendations for healthy lifestyles and healthy behavior (physical activity, nutrition, stress, addictive substances) during pregnancy, and 79.0% preferred these media contents during lactation. Movement exercises, relaxation exercises, and recommendations for the avoidance of specific medications during pregnancy were of interest for more than half of the surveyed women (Table 4 and Table 5).

In multivariate logistic regression analyses, we investigated women’s preferred media content during pregnancy and lactation in association with independent variables (Table 4 and Table 5). We found non-pregnant women, as compared to pregnant women, to have a significantly higher probability to prefer media content regarding potential alternatives to tobacco and alcohol during lactation (OR 3.09, 95% CI 1.20, 7.90), on recommendations for the avoidance of specific medications during pregnancy (OR 3.29, 95% CI 1.24, 8.71), and recommendations of essential supplements during pregnancy (OR 2.43, 95% CI 1.07, 5.53). Non-pregnant women had a lower probability of preferring media content regarding potential alternatives to tobacco and alcohol during pregnancy (OR 0.33, 95% CI 0.13, 0.86) as compared to pregnant women. Women who were not currently lactating, as compared to lactating women, were significantly more likely to prefer media content regarding cooking recipes during lactation (OR 4.61, 95% CI 1.44, 14.74). Women who were not in a firm relationship had a lower probability of preferring media content on movement exercises during lactation (OR 0.15, 95% CI 0.03, 0.82) as compared to women in a firm relationship. The probability of preferring media content regarding recommendations for healthy lifestyles/behavior during lactation increased with decreasing age (OR 0.91, 95% CI 0.84, 0.99).

Women with a lower SES, as compared to high-SES women, were less likely to prefer media content regarding recommendations for a healthy lifestyle/behavior during lactation (OR 0.17, 95% CI 0.04, 0.78), on recommendations for the avoidance of lifestyle-related risks/behaviors during pregnancy (OR 0.24, 95% CI 0.06, 0.90), and on potential alternatives to tobacco and alcohol during pregnancy (OR 0.19, 95% CI 0.04, 0.92). However, compared to high SES women, women with a lower SES had a higher probability of preferring recipes for cooking during pregnancy (OR 3.67, 95% CI 1.02, 13.26) and lactation (OR 6.30, 95% CI 1.54, 25.84) and recommendations for the avoidance of specific medications during pregnancy (OR 6.55, 95% CI 1.28, 33.44). Women with a middle SES as compared to high SES women had a higher probability of preferring media content regarding movement exercises during pregnancy (OR 3.45, 95% CI 1.64, 7.25) and recommendations for the avoidance of specific medications during pregnancy (OR 3.03, 95% CI 1.09, 8.43). Non-German women, compared to German women, had a lower probability of preferring media content on recommendations for healthy lifestyles/behavior during pregnancy (OR 0.31, 95% CI 0.11, 0.83), on potential alternatives to tobacco and alcohol during lactation (OR 0.21, OR 0.04, 0.96), and on recommendations for the avoidance of specific medications during pregnancy (OR 0.12, 95% CI 0.01, 0.93). Non-German women, compared to German women, had a much higher probability of preferring media content regarding potential alternatives to tobacco and alcohol during pregnancy (OR 8.97, 95% CI 1.12, 71.54). Women with statutory health insurance, as compared to women with private health insurance, had a lower probability of preferring media content regarding movement exercises during pregnancy (OR 0.25, 95% CI 0.08, 0.77) and on recommendations of essential supplements during pregnancy (OR 0.25, 95% CI 0.08, 0.77). Women who did not have private or statutory health insurance had a significantly lower probability of preferring media content including information regarding potential lifestyle-related risks during lactation (OR 0.09, 95% CI 0.01, 0.91) and on recommendations of essential supplements during pregnancy (OR 0.09, 95% CI 0.02, 0.50).

## 4. Discussion

### 4.1. Key Results

In the setting of this study, most women used media regarding health-related behaviors during pregnancy and lactation. Fertility status (currently pregnant and currently lactating) and the women’s age were significantly associated with media use regarding health-related behaviors during pregnancy and lactation. Older women, pregnant women, and lactating women had a higher probability of using media during pregnancy and lactation, respectively. Most women preferred a combination of analog and digital media in the gynecological and obstetric care settings and at home. Women would like to see information brochures and flyers on health-related behaviors during pregnancy and lactation for use in gynecological and obstetric care settings, and for media use at home, they would like to have books. The probability of preferring the most popular media formats in gynecological and obstetric care settings and at home was associated with pregnancy status, relationship status, SES, ethnicity, and health insurance status. Pregnant women had a stronger preference for the use of an app at home. Non-pregnant women had a stronger preference for information brochures in obstetric care. Most women said that media content should focus on recommendations for health-related behaviors during pregnancy and lactation. All potential influence factors were associated with the probability of preferring a specific media content. The SES seemed to play a major role in the probability of preferring a specific media content, followed by pregnancy status, ethnicity, and health insurance status. Pregnant women were more likely to say that media content should focus on potential alternatives to tobacco and alcohol during pregnancy. Non-pregnant women had stronger preferences for recipes for cooking, potential alternatives to tobacco and alcohol during lactation, and for recommendations for the avoidance of specific medications and supplements during pregnancy.

### 4.2. Discussion of the Key Results

Most of the women in our study used media regarding health-related behaviors during pregnancy and lactation. We found fertility status (pregnant and lactating) and the women’s age to be significantly associated with media use regarding health-related behaviors during pregnancy and lactation, respectively. In fact, we expected an increased media use regarding health-related behaviors during pregnancy when women were pregnant or lactating. This demonstrates that women use media when the specific event occurs. However, this in turn also implies that preconceptionally usage of media regarding health-related behaviors during pregnancy and lactation is less likely. As most women consume alcohol during pregnancy, specifically in the first trimester, there is a need to address women preconceptionally [23]. Our study shows that media use regarding health-behaviors during pregnancy and lactation increased with increasing age. Therefore, future analog and digital media should be designed with contents and formats according to the target group of younger women of childbearing age.

Combining analog and digital media for use in gynecological and obstetric care meets women’s preferences. In contrast to our expectations, digital media were less preferred as compared to analog media regarding health-related behaviors during pregnancy and lactation. A reason for the preference for analog media might be a current overload of existing mobile apps and other digital media types regarding health-related behaviors [24,25]. Combining analog (or real) and digital environments was a preferred approach as a personal approach to sensitive healthcare topics such as pregnancy and lactation are urgently needed [25,26]. In our study, information brochures and flyers were the most preferred media types regarding health-related behaviors during pregnancy and lactation in gynecological and obstetric care, and books, addresses, and apps were preferred for use at home. For use at home, we found pregnant women to have a strong preference for mobile apps regarding health-related behaviors during pregnancy and lactation. A recent ethnographic study found that patients consider mobile health apps to be a useful complementary tool [27]. However, the patients faced some major problems and uncertainties, such as questions on the optimal usage and the price of specific apps; they doubted the validity of the information delivered and questioned issues of security and privacy [27]. Studies found that mobile apps can give conflicting information, and the source of recommendation is often unclear and insufficient, and therefore they are often used as complementary to evidence-based information [28,29]. Therefore, women’s perception regarding books might be more valid and evidence-based as compared to preconceptions regarding mobile apps [27]. Pregnancy is a sensitive period for the unborn child, and, therefore, sensitive strategies and motivational approaches are of urgent need in gynecological and obstetric care. Flyers and brochures at a gynecologist or midwife practice are intended to show the most relevant information on a specific topic in a concise format [30,31,32]. Moreover, we found younger women and German women to be more likely to prefer information brochures on health-related behaviors during pregnancy and lactation in gynecological and obstetric care. An explanation might be that older women may have already experienced a pregnancy, and thus feel more self-confident and less uncertain as compared to younger women [33]. Reaching immigrants and ethnic minorities with media-based interventions is often difficult due to language barriers, decreased utilization of medical care, and lack of knowledge of existing preventive measures and available health care services [34,35]. Women with statutory health insurance were more likely to prefer mobile apps and relevant addresses for use at home as compared to women with private health insurance. A reason for this might be that women with private health insurance benefit from faster, easier, and higher quality access to medical care and treatment (also due to higher accounting rates) [36].

In our study, we found that most women said that information about pregnancy and lactation should focus on recommendations regarding health-related behaviors during pregnancy and lactation. People prefer valid, evidence-based and reliable information and recommendations, specifically in phases of personal uncertainties and in new, non-daily situations [31,37]. We found associations between all potential influence factors and the women’s preferred media content. All potential influence factors were associated with the probability of preferring a specific media content during pregnancy and lactation. The results provide a valuable basis for tailored interventions. Pregnant women were more likely to say that information should focus on potential alternatives to tobacco and alcohol during pregnancy as compared to non-pregnant women. From the perspective of pregnant women, information on potential alternatives to tobacco and alcohol during pregnancy seemed to be relevant. It might be well that there exists a lack of media content regarding potential alternatives to tobacco and alcohol during pregnancy as compared to more popular topics such as recipes for cooking or movement exercises, as covered by a variety of mobile apps, social networks, and other media formats. We found non-pregnant women, as compared to pregnant women, to have stronger preferences for recipes for cooking, potential alternatives to tobacco and alcohol during lactation, and for recommendations for the avoidance of specific medications and supplements during pregnancy. Pregnant women might have a knowledge advantage and read much more extensively than non-pregnant women. These results demonstrate how to reach women in the pre-conception period. As it is of urgent need to initiate preventive measures for LRRFs in the pre-conception period so as to avoid and reduce risks from the very beginning of a pregnancy, women should be addressed with tailored media content that addresses their interests before conception. Moreover, the women’s SES was associated with a variety of assessed media contents. This study shows that there are also topics of interest among low-SES women, e.g., low-SES women were more likely to be reached with recipes for cooking during pregnancy and lactation and recommendations for the avoidance of specific medications, but they were less likely to be reached with recommendations for the avoidance of lifestyle-related risks/behaviors during pregnancy or recommendations for potential alternatives to tobacco and alcohol. To address the relevant target groups, it is necessary to tailor the content of the intervention to the desires and preferences of the respective target group [38,39]. Low SES women are difficult to reach, and, therefore, it is important to utilize strategies to maximize the probability of also reaching those who are difficult to reach. This group is of high interest and importance in the health care system as they cause high costs to the system. Similarly, a precondition to successfully tailoring preventive measures for immigrants and ethnic minority groups is to understand the needs and desires of the target group. The non-German women in our study were almost nine times more likely to prefer media content regarding potential alternatives to tobacco and alcohol. Depending on the country of origin, tobacco smoking or alcohol consumption are attached to cultural norms and social attitudes. For example, in Russia, toasting and drinking are major communication rituals [40], and, accordingly, a study in Russia found that 60.0% of pregnant women drank although pregnancy was known [41]. Therefore, addressing target groups with tailored content is of urgent need.

As compared to women with private health insurance, women with statutory health insurance or others had significantly lower probabilities of preferring specific media contents regarding health-related behaviors during pregnancy and lactation, such as movement exercises during pregnancy or recommendations of essential supplements. To the best of our knowledge, there are no studies investigating differences in attitudes, knowledge, and behaviors among clients across health insurances that would allow placing these results in context. Further studies might provide deeper insights into these differences.

Our research results provide relevant preliminary information on the preferred media types and contents across different target groups, enabling the development of tailored preventive measures.

### 4.3. Evaluation of Potential Limitations

Our study has some potential limitations that need to be addressed. First, the sample size in our study was small, and the setting was limited to the federal state of Baden-Wuerttemberg. Therefore, our results need to be interpreted cautiously and might not similarly apply to other states or countries. However, we discussed each result in detail and placed, when possible, our results in the context of the current state of the art. Second, as compared to the general population, the socioeconomic status was comparably high in our study population, with 46.2% in the high and 41.8% in the middle socioeconomic status group. Women from the lower socioeconomic status group were underrepresented, and women in the higher social status group were overrepresented in our study. In the general German population, 26.6% have a low social status, and 24.5% have a high socioeconomic status [42]. An explanation might be that we recruited women in different settings of obstetric and gynecological care during non-obligatory appointments, e.g., during a tour of a delivery and maternity ward, prenatal classes, and parents’ evenings in hospitals. Studies have found that women from the lower socioeconomic status group have lower compliance with antenatal care, e.g., a higher socioeconomic status is positively associated with knowledge and practice of antenatal care [43], and late attenders, in comparison to early attenders, of antenatal care are more likely to be of lower socioeconomic status [44]. Therefore, the results of our study might not fully apply to the general German population, and they may not be transferable to other countries. Third, to address our specific research questions, we could not use a validated questionnaire, and therefore applied a self-developed questionnaire. However, we pilot-tested our questionnaire with 12 women and reached saturation after six pilot tests.

Despite these limitations, our study provides relevant information for the development of educational interventions on LRRFs in gynecological and obstetric care settings. Our results allow the design of tailored media-based interventions by focusing on target groups with regard to media format and content. The results of our study should be implemented in educational interventions and evaluated for efficacy regarding health knowledge on LRRFs and subsequent behavioral changes.

## 5. Conclusions

This study has demonstrated that about three-quarters of the surveyed women in Germany use media during pregnancy and lactation. These were specifically pregnant and lactating women and comparably older women. Our study results have shown that non-pregnant women have stronger preferences for a variety of media contents as compared to pregnant women. From the perspective of pregnant women, media content should focus on rarely addressed issues, such as potential alternatives to tobacco and alcohol. The results imply that there are diverse topics on health-related behaviors during pregnancy and lactation that are of interest to non-pregnant women, and, thus, reaching women in the pre-conception period—and this applies also to pregnant and lactating women—might be successful when preventive interventions are tailored to the women’s requirements combined with information that is rarely addressed; however, acceptance may vary across personal attributes. Our study suggests that in the context of health-related behaviors during pregnancy and lactation, besides fertility status, a women’s SES, ethnicity, and health insurance status in particular determine which content is of interest. Future educational interventions should be tailored to the target group and include digital and analog components covering recommendations for a healthy lifestyle and behavior (physical activity, nutrition, stress, and consumption of addictive substances), e.g., information brochures and flyers in gynecological and obstetric care settings accompanied by a mobile app for use at home.

## Figures and Tables

**Figure 1 ijerph-18-09840-f001:**
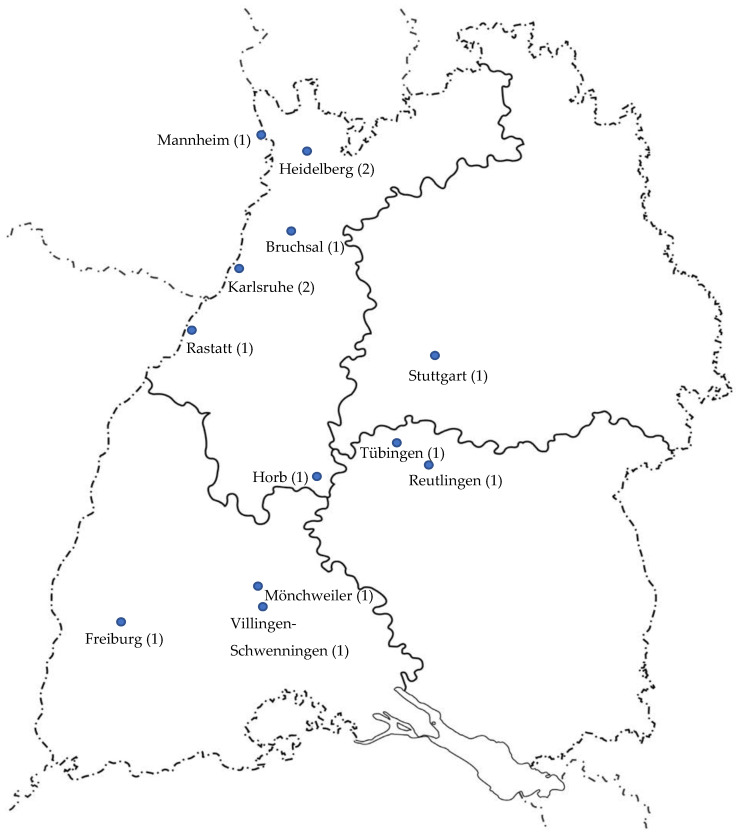
Study’s sample points.

**Figure 2 ijerph-18-09840-f002:**
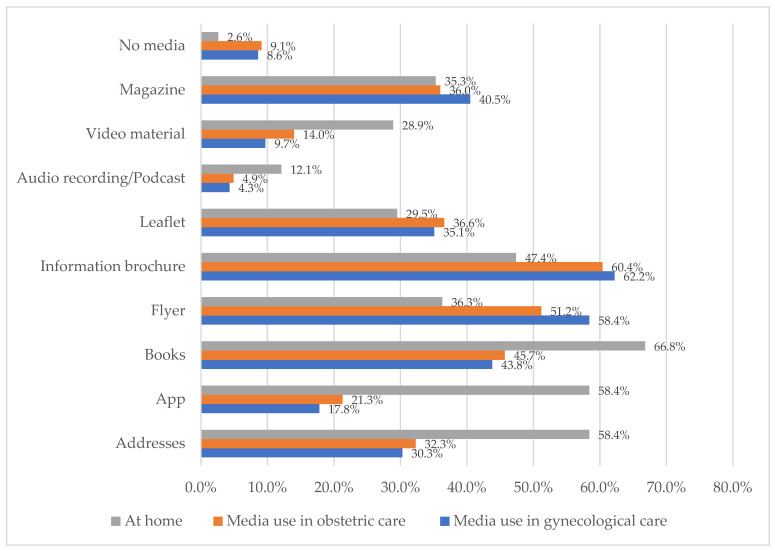
Women’s preferred media formats regarding health-related behaviors during pregnancy and lactation in gynecological and obstetric care (n = 219).

**Table 1 ijerph-18-09840-t001:** Sample characteristics with frequencies (%) and means (SD).

Characteristics (N = 219)	Total
	% (n)/M (SD)
Currently pregnant (yes, in %)	57.8% (n = 126)
Currently lactating (yes, in %))	17.1% (n = 37)
Firm relationship (yes, in %)	91.7% (n = 176)
Age (M (SD))	30.97 (5.516)
Socioeconomic status	
High (in %)	46.2% (n = 84)
Middle (in %)	41.8% (n = 76)
Low (in %)	12.1% (n = 22)
Ethnicity	
German (in %)	81.5% (n = 150)
Others (in %)	18.5% (n = 34)
Health insurance status	
Private (in %)	14.8% (n = 27)
Statutory (in %)	79.1% (n = 144)
Others/None (in %)	6.0% (n = 11)

Note: Data were missing for currently pregnant (n = 1), currently lactating (n = 2), firm relationship (n = 27), age (n = 26), socioeconomic status (n = 37), ethnicity (n = 35), and health insurance status (n = 37). Values are percentages for categorical factors or means (with standard deviations) for continuous factors.

**Table 2 ijerph-18-09840-t002:** Media use regarding health-related behaviors during pregnancy and lactation and its association with independent variables, expressed in odds ratios (95% confidence intervals).

Independent Variables	Media Use Regarding Health-Related Behaviors during Pregnancy			Media Use Regarding Health-Related Behaviors during Lactation		
	% or M (SD)		OR (95% CI)	% or M (SD)		OR (95% CI)
	Media Use (n = 157)	No Media Use (n = 43)	(n = 166)	Media Use (n = 157)	No Media Use (n = 43)	(n = 166)
Currently pregnant						
Yes	98.1%	1.9%	1			
No	76.5%	23.5%	0.30 (0.13, 0.67) **			
Currently lactating						
Yes				84.4%	15.6%	1
No				71.6%	28.4%	0.21 (0.05, 0.98) *
Firm relationship						
Yes	89.7%	10.3%	1	73.9%	26.1%	1
No	66.7%	33.3%	0.53 (0.13, 2.10)	58.3%	41.7%	0.77 (0.19, 3.03)
Age	31.8 (4.8)	28.6 (5.9)	1.11 (1.02, 1.21) *	31.9 (4.7)	29.3 (5.7)	1.08 (1.00, 1.17) *
Socioeconomic status						
High	91.7%	8.3%	1	74.7%	25.3%	1
Middle	86.7%	13.3%	0.84 (0.35, 1.99)	71.4%	28.6%	1.02 (0.46, 2.27)
Low	82.4%	17.6%	1.95 (0.39, 9.86)	66.7%	33.3%	0.97 (0.22, 4.28)
Ethnicity						
German	86.8%	13.2%	1	72.9%	27.1%	1
Non-German	93.1%	6.9%	1.29 (0.44, 3.79)	68.8%	31.2%	0.80 (0.32, 2.00)
Health insurance						
Private	95.2%	4.8%	1	70.8%	29.2%	1
Statutory	86.9%	13.1%	1.24 (0.38, 4.01)	73.2%	26.8%	1.42 (0.46, 4.38)
Others/None	85.7%	14.3%	0.38 (0.07; 2.11)	60.9%	40.0%	0.51 (0.09, 2.83)
Nagelkerkes R^2^			0.195			0.122

Note: Significance key: * *p* ≤ 0.05 and ** *p* ≤ 0.01; M = mean, SD = standard deviation, OR = odds ratios; CI = confidence interval.

**Table 3 ijerph-18-09840-t003:** Probability of preferring selected media formats on health-related behaviors during pregnancy and lactation in gynecological and obstetric care and for use at home in association with independent variables, expressed in odds ratios (95% confidence intervals).

Independent Variables	Media in Gynecological Care		Media in Obstetric Care		Media at Home		
	Information Brochures (n = 163)	Flyers (n = 163)	Information Brochures (n = 168)	Flyers (n = 145)	Books (n = 167)	Addresses (n = 167)	Apps (n = 167)
Currently pregnant							
Yes	1	1	1	1	1	1	1
No	1.85 (0.77, 4.41)	1.43 (0.63, 3.25)	3.26 (1.29, 8.20) **	0.98 (0.42, 2.28)	1.24 (0.54, 2.81)	0.81 (0.36, 1.83)	0.44 (0.20, 1.00) *
Currently lactating							
Yes	1	1	1	1	1	1	1
No	1.60 (0.49, 5.24)	0.96 (0.31, 2.98)	2.76 (0.83, 9.22)	0.57 (0.18, 1.79)	1.35 (0.43, 4.21)	0.83 (0.27, 2.53)	0.54 (0.17, 1.65)
Firm relationship							
Yes	1	1	1	1	1	1	1
No	0.43 (0.11, 1.66)	1.83 (0.44, 7.53)	0.23 (0.06, 0.95) *	1.61 (0.38, 6.78)	1.69 (0.43, 6.55)	1.06 (0.30, 3.78)	1.51 (0.42, 5.48)
Age	0.92 (0.86, 0.99) *	0.95 (0.89, 1.02)	0.91 (0.85, 0.97) **	0.96 (0.89, 1.03)	1.00 (0.94, 1.06)	0.95 (0.89, 1.01)	0.96 (0.90, 1.03)
Socioeconomic status							
High	1	1	1	1	1	1	1
Middle	0.84 (0.41, 1.75)	1.71 (0.84, 3.48)	0.66 (0.32, 1.37)	2.21 (1.05, 4.66) *	0.54 (0.26, 1.12)	0.29 (0.13, 0.61) ***	0.81 (0.39, 1.66)
Low	0.36 (0.10, 1.29)	2.08 (0.59, 7.31)	0.47 (0.13, 1.72)	0.48 (0.11, 2.00)	0.43 (0.13, 1.45)	0.20 (0.06, 0.70) **	0.58 (0.16, 2.02)
Ethnicity							
German	1	1	1	1	1	1	1
Non-German	0.42 (0.18, 0.99) *	1.16 (0.49, 2.75)	0.38 (0.16, 0.89) *	0.71 (0.28, 1.76)	0.86 (0.37, 2.02)	1.80 (0.72, 4.49)	0.28 (0.12, 0.67) **
Health insurance							
Private	1	1	1	1	1	1	1
Statutory	0.51 (0.17, 1.48)	2.00 (0.73, 5.46)	0.90 (0.34, 2.42)	0.57 (0.20, 1.60)	0.97 (0.36, 2.66)	2.72 (1.00, 7.45) *	2.73 (1.03, 7.22) *
Others/None	0.16 (0.32, 0.78) *	4.34 (0.92, 20.42)	0.28 (0.06, 1.36)	0.49 (0.11, 2.25)	2.08 (0.35, 12.49)	0.60 (0.13, 2.88)	2.31 (0.50, 10.61)
Nagelkerkes R^2^	0.139	0.102	0.186	0.105	0.047	0.165	0.134

Note: Significance key: * *p* ≤ 0.05, ** *p* ≤ 0.01, and *** *p* ≤ 0.001; values are odds ratios with 95% confidence intervals in parentheses.

**Table 4 ijerph-18-09840-t004:** Women’s preferred media content during pregnancy and lactation in association with independent variables, expressed in odds ratios (95% confidence intervals).

Independent Variables	Recommendations for Healthy Lifestyle/Behaviors		Recommendations for Avoidance of Lifestyle-Related Risks/Behaviors		Information on Potential Lifestyle-Related Risks		Recipes for Cooking	
	During Pregnancy (n = 167)	During Lactation (n = 159)	During Pregnancy (n = 167)	During Lactation (n = 159)	During Pregnancy (n = 167)	During Lactation (n = 159)	During Pregnancy (n = 167)	During Lactation (n = 159)
% (n) of women who prefer media content	80.1% (n = 149)	79.0% (n = 139)	47.3% (n = 88)	47.7% (n = 84)	46.2% (n = 86)	43.2% (n = 76)	33.3% (n = 62)	43.8% (n = 77)
Currently pregnant								
Yes	1	1	1	1	1	1	1	1
No	1.18 (0.41, 3.44)	0.66 (0.25, 1.78)	0.85 (0.39, 1.86)	1.01 (0.46, 2.23)	0.86 (0.40, 1.85)	0.67 (0.30, 1.50)	1.47 (0.67, 3.24)	1.68 (0.75, 3.81)
Currently lactating								
Yes	1	1	1	1	1	1	1	1
No	2.93 (0.84, 10.24)	1.51 (0.45, 5.03)	0.54 (0.19, 1.54)	0.98 (0.35, 2.76)	2.08 (0.72, 6.02)	1.47 (0.49, 4.40)	2.31 (0.75, 7.15)	4.61 (1.44, 14.74) **
Firm relationship								
Yes	1	1	1	1	1	1	1	1
No	^a^	1.74 (0.33, 9.34)	2.18 (0.61, 7.81)	0.85 (0.25, 2.95)	1.07 (0.31, 3.71)	1.03 (0.29, 3.67)	0.33 (0.08, 1.33)	0.34 (0.09, 1.30)
Age	0.93 (0.85, 1.20)	0.91 (0.84, 0.99) *	0.99 (0.93, 1.06)	1.01 (0.94, 1.08)	0.97 (0.91, 1.03)	1.01 (0.94, 1.07)	1.00 (0.94, 1.07)	1.06 (0.99, 1.14)
Socioeconomic status								
High	1	1	1	1	1	1	1	1
Middle	0.81 (0.34, 1.94)	0.91 (0.37, 2.24)	0.58 (0.29, 1.17)	0.82 (0.41, 1.64)	1.24 (0.62, 2.47)	1.07 (0.52, 2.18)	1.74 (0.83, 3.64)	1.67 (0.81, 3.46)
Low	0.98 (0.16, 6.23)	0.17 (0.04, 0.78) *	0.24 (0.06, 0.90) *	1.09 (0.31, 3.89)	0.96 (0.28, 3.32)	1.58 (0.43, 5.71)	3.67 (1.02, 13.26) *	6.30 (1.54, 25.84) **
Ethnicity								
German	1	1	1	1	1	1	1	1
Non-German	0.31 (0.11, 0.83) *	0.53 (0.19, 1.49)	0.64 (0.27, 1.47)	0.49 (0.21, 1.18)	0.58 (0.25, 1.37	0.54 (0.22, 1.31)	1.07 (0.44, 2.57)	0.86 (0.35, 2.12)
Health insurance								
Private	1	1	1	1	1	1	1	1
Statutory	0.16 (0.02, 1.28)	0.37 (0.08, 1.80)	1.27 (0.49, 3.28)	1.11 (0.42, 2.92)	0.53 (0.20, 1.39)	0.61 (0.23, 1.64)	0.92 (0.33, 2.56)	0.90 (0.33, 2.52)
Others/None	0.60 (0.03, 11.20)	0.39 (0.04, 3.48)	0.19 (0.03, 1.13)	0.25 (0.04, 1.49)	0.24 (0.05, 1.21)	0.09 (0.01, 0.91) *	0.86 (0.17, 4.42)	0.67 (0.12, 3.77)
Nagelkerkes R^2^	0.217	0.136	0.103	0.052	0.078	0.093	0.072	0.134

Note: Significance key: * *p* ≤ 0.05 and ** *p* ≤ 0.01; values are odds ratios with 95% confidence intervals in parentheses. ^a^ no cases.

**Table 5 ijerph-18-09840-t005:** Women’s preferred media content during pregnancy and lactation in association with independent variables, expressed in odds ratios (95% confidence intervals).

Independent Variables	Movement Exercises		Relaxation Exercises		Potential Alternatives to Tobacco and Alcohol		Recommendations for Avoidance of Specific Medications		Recommendations for Essential Supplements	
	During Pregnancy (n = 167)	During Lactation (n = 159)	During Pregnancy (n = 167)	During Lactation (n = 159)	During Pregnancy (n = 168)	During Lactation (n = 159)	During Pregnancy (n = 167)	During Lactation (n = 159)	During Pregnancy (n = 167)	During Lactation (n = 159)
% (n) of women who prefer media content	59.7% (n = 111)	40.3% (n = 71)	55.4% (n = 103)	46.6% (n = 82)	17.1% (n = 33)	21.0% (n = 37)	59.1% (n = 110)	58.0% (n = 102)	48.4% (n = 90)	47.2% (n = 83)
Currently pregnant										
Yes	1	1	1	1	1	1	1	1	1	1
No	0.86 (0.38, 1.95)	1.07 (0.48, 2.39)	1.19 (0.55, 2.57)	1.44 (0.66, 3.16)	0.33 (0.13, 0.86) *	3.09 (1.20, 7.90) *	3.29 (1.24, 8.71) *	1.62 (0.71, 3.69)	2.43 (1.07, 5.53) *	1.80 (0.80, 4.06)
Currently lactating										
Yes	1	1	1	1	1	1	1	1	1	1
No	1.49 (0.51, 4.36)	1.15 (0.41, 3.25)	1.56 (0.55, 4.42)	0.86 (0.31, 2.41)	0.81 (0.24, 2.74)	1.33 (0.41, 4.37)	1.19 (0.35, 4.10)	1.13 (0.39, 3.27)	1.89 (0.64, 5.63)	1.38 (0.48, 3.97)
Firm relationship										
Yes	1	1	1	1	1	1	1	1	1	1
No	0.47 (0.13, 1.70)	0.15 (0.03, 0.82) *	0.95 (0.27, 3.31)	0.41 (0.12, 1.49)	3.01 (0.55, 16.33)	0.42 (0.08, 2.13)	0.33 (0.06, 1.80)	0.52 (0.14, 1.96)	0.45 (0.12, 1.62)	0.48 (0.13, 1.76)
Age	0.99 (0.93, 1.06)	1.06 (0.98, 1.13)	1.01 (0.95, 1.08)	1.01 (0.95, 1.08)	1.07 (0.98, 1.16)	0.92 (0.84, 1.01)	0.94 (0.86, 1.03)	0.95 (0.89, 1.02)	0.94 (0.88, 1.01)	1.00 (0.94, 1.07)
Socioeconomic status										
High	1	1	1	1	1	1	1	1	1	1
Middle	3.45 (1.64, 7.25) ***	1.35 (0.66, 2.75)	0.61 (0.30, 1.21)	0.89 (0.44, 1.78)	0.39 (0.15, 1.04)	1.28 (0.53, 3.09)	3.03 (1.09, 8.43) *	0.80 (0.40, 1.61)	0.64 (0.31, 1.30)	0.70 (0.34, 1.43)
Low	2.57 (0.71, 9.24)	2.52 (0.63, 10.01)	0.61 (0.18, 2.10)	1.18 (0.33, 4.22)	0.19 (0.04, 0.92) *	1.01 (0.19, 5.23)	6.55 (1.28, 33.44) *	2.61 (0.58, 11.68)	0.47 (0.13, 1.72)	0.98 (0.27, 3.54)
Ethnicity										
German	1	1	1	1	1	1	1	1	1	1
Non-German	1.04 (0.44, 2.46)	0.92 (0.37, 2.25)	1.46 (0.64, 3.34)	0.86 (0.36, 2.03)	8.97 (1.12, 71.54) *	0.21 (0.04, 0.96) *	0.12 (0.01, 0.93) *	0.82 (0.34, 1.97)	0.55 (0.23, 1.29)	0.43 (0.17, 1.08)
Health insurance										
Private	1	1	1	1	1	1	1	1	1	1
Statutory	0.25 (0.08, 0.77) *	0.86 (0.31, 2.33)	1.98 (0.72, 5.43)	0.90 (0.34, 2.39)	1.76 (0.52, 5.94)	1.67 (0.43, 6.50)	0.51 (0.15, 1.76)	0.78 (0.28, 2.12)	0.25 (0.08, 0.77) *	0.37 (0.13, 1.06)
Others/None	0.20 (0.04, 1.02)	0.53 (0.10, 2.78)	4.21 (0.90, 19.68)	0.38 (0.07, 2.02)	0.69 (0.12, 4.04)	2.88 (0.42, 19.70)	1.48 (0.25, 8.92)	0.77 (0.15, 3.88)	0.09 (0.02, 0.50) **	0.44 (0.08, 2.33)
Nagelkerkes R^2^	0.143	0.082	0.057	0.041	0.232	0.175	0.252	0.065	0.158	0.100

Note: Significance key: * *p* ≤ 0.05, ** *p* ≤ 0.01, and *** *p* ≤ 0.001; values are odds ratios with 95% confidence intervals in parentheses.

## Data Availability

The data presented in this study are available on request from the corresponding author. The data are not publicly available due to ongoing studies.

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
