# Peer review of "Women’s Media Use and Preferences of Media-Based Interventions on Lifestyle-Related Risk Factors in Gynecological and Obstetric Care: A Cross-Sectional Multi-Center Study in Germany"

_ijerph, 2021, doi:10.3390/ijerph18189840_

Round 1

Reviewer 1 Report

Thank you for your submission. 

This is an interesting paper to read. The topic has high relevance. The authors made great work in terms of methodology and the paper sounds scientific and well written.

Author Response

Dear reviewer,

we wish to thank you for your valuable feedback and the positive evaluation of our manuscript. We highly appreciate the time you put into reviewing our manuscript and writing the report.

Kind regards,

Manuela Bombana, on behalf of all co-authors

Reviewer 2 Report

Thank you for this study. The findings are fascinating and can provide valuable insight for health educators. 

Did you conduct a power analysis for the sample size?

You acknowledged that the survey you used was not validated, but it does appear to have face validity. 

Would you consider a table of topics the participants were most interested in based on situation (rather than setting), i.e. pregnant, breastfeeding, gynecology patients.

Did you separate postpartum breastfeeding and postpartum not breastfeeding in the data analysis?

Author Response

Dear reviewer,

we wish to thank you for your valuable feedback and the positive evaluation of our manuscript. We highly appreciate the time you put into reviewing our manuscript and writing the report.

Please find our point-by-point responses in the following paragraph:

1. Did you conduct a power analysis for the sample size?

Response 1: Because of the explorative nature of the study – no hypotheses were stated – we did not perform an explicit power analysis.

But a post hoc power analysis showed:

A logistic regression of a binary response variable (Y) on a binary independent variable (X) with a sample size of 200 observations (of which 50% are in the group X=0 and 50% are in the group X=1) achieves 86% power at a 0,050 significance level to detect a change in Prob(Y=1) from the baseline value of 0,700 to 0,875. This change corresponds to an odds ratio of 3,000. An adjustment was made since a multiple regression of the independent variable of interest on the other independent variables in the logistic regression obtained an R-Squared of 0,010.

2. You acknowledged that the survey you used was not validated, but it does appear to have face validity. 

Response 2: We created a focus group with experts (gynecologists and midwives) and discussed with the group members the plausibility of every single question for the concept it aims to measure. We believe that the questionnaire is logical valid as captured by face validity.

3. Would you consider a table of topics the participants were most interested in based on situation (rather than setting), i.e. pregnant, breastfeeding, gynecology patients.

Response 3: We believe that in the context of our study and our planned intervention, it is of relevance to see the topics with regard to its setting. We also find the reviewers suggestions interesting and may consider addressing these in a future study.

4. Did you separate postpartum breastfeeding and postpartum not breastfeeding in the data analysis?

Response 4: We distinguished postpartum breastfeeding from postpartum not breastfeeding by the application of the dichotomous variable “currently lactating” (yes/no).

Kind regards,

Manuela Bombana, on behalf of all co-authors

Reviewer 3 Report

In my opinion the topic of the study is very intersting. I would change the translation becasue in some cases the syntax is hard to understand.  I would change ,,old" positions  in the bibliography: 7,14,18,20,21,32,41,44.

Author Response

Dear reviewer,

we wish to thank you for your valuable feedback and the positive evaluation of our manuscript. We highly appreciate the time you put into reviewing our manuscript and writing the report.

We followed your suggestion and removed the translation of references and checked all references and removed - when possible and plausible - older references and replaced them by newer ones.

Kind regards,

Manuela Bombana, on behalf of all co-authors